# A low rate of end-stage kidney disease in membranous nephropathy: A single centre study over 2 decades

Joshua Storrar [1,2]*, Tarra Gill-Taylor[1], Rajkumar Chinnadurai[1], Constantina Chrysochou[1], Dimitrios Poulikakos[1], Francesco Rainone[1], James Ritchie[1], Elizabeth Lamerton[1], Philip A. Kalra[1], Smeeta Sinha[1]

1 Salford Royal Hospital, Northern Care Alliance NHS Foundation Trust, Salford, United Kingdom,
2 University of Manchester, Manchester, United Kingdom

* joshua.storrar2@nca.nhs.uk

## Abstract

### Introduction

Membranous nephropathy is the commonest cause of nephrotic syndrome in non-diabetic Caucasian adults over the age of 40 years. Primary membranous nephropathy is limited to the kidneys. Clinical management aims to induce remission, either spontaneously with supportive care, or with immunosuppression. Here, we describe the natural history of this condition in a large tertiary centre in the UK.

### Methods

178 patients with primary membranous nephropathy were identified over 2 decades. We collected data on demographics, baseline laboratory values, treatment received and outcomes including progression to renal replacement therapy and death. Analysis was performed on the whole cohort and specific subgroups. Univariate and multivariate Cox regression was also performed.

### Results

Median age was 58.3 years with 63.5% male. Median baseline creatinine was 90μmol/L and urine protein-creatinine ratio 664g/mol. Remission (partial or complete) was achieved in 134 (75.3%), either spontaneous in 60 (33.7%) or after treatment with immunosuppression in 74 (41.6%), and of these 57 (42.5%) relapsed. Progression to renal replacement therapy was seen in 10.1% (much lower than classically reported) with mortality in 29.8%. Amongst the whole cohort, those who went into remission had improved outcomes compared to those who did not go into remission (less progression to renal replacement therapy [4.5% vs 28%] and death [20.1% vs 67%]). Those classified as high-risk (based on parameters including eGFR, proteinuria, serum albumin, PLA2R antibody level, rate of renal function decline) also had worse outcomes than those at low-risk (mortality seen in 52.6% vs 10.8%, p<0.001). The median follow-up period was 59.5 months.

**Data Availability Statement:** All relevant data are within the manuscript and its Supporting Information files.

**Funding:** The Salford glomerulonephritis research group was generously supported by an unrestricted project grant from Vifor. The funders had no role in study design, data collection and analysis, decision to publish, or preparation of the manuscript.

**Competing interests:** S Sinha has received grants from Johnson and Johnson and AstraZeneca; speaker and lecture fees from AstraZeneca, Napp, Bayer, Sanofi-Genzyme, Novartis, and Vifor Pharma; is on advisory boards for Novartis, AstraZeneca, Bayer, and Travere; and has a clinical consultancy role with Sanifit. PA Kalra has received grants from Astellas, Vifor Pharma, BergenBio and Evotec; speaker and lecture fees from AstraZeneca, Napp, Bayer, Novartis, Vifor Pharma, Pharmacosmos, Boehringer Ingelheim; is on advisory boards for AstraZeneca and Vifor Pharma; and has a consultancy role with Bayer, Astella, Otsuka and Unicyte. This does not alter our adherence to PLOS ONE policies on sharing data and materials.

## Conclusion

We provide a comprehensive epidemiologic analysis of primary membranous nephropathy at a large tertiary UK centre. Only 10.1% progressed to renal replacement therapy. For novelty, the KDIGO risk classification was linked to outcomes, highlighting the utility of this classification system for identifying patients most likely to progress.

## Introduction

Membranous nephropathy (MN) is the most common cause of nephrotic syndrome (NS) in non-diabetic Caucasian adults over the age of 40 years, with an incidence of between 8–10 cases per 1 million [1]. It classically presents with proteinuria greater than 3.5g/day, hypoalbuminaemia, oedema and hyperlipidaemia [2]. MN can be classified as either primary or secondary. Primary MN (PMN), seen in 80% of cases, is an autoimmune condition in which pathology is limited to the kidney, whereas secondary MN, accounting for 20% of cases, is associated with other diseases [3]. The pathogenesis of PMN is mediated by the production of antibodies against antigens that are expressed on the glomerular podocytes. In humans the main target antigen that has been identified is the M-type phospholipase A2 receptor (PLA2R), with 70% of patients producing autoantibodies directed against it [4], with a further 1–3% producing antibodies against thrombospondin type-1 domain-containing 7a (THSD7A) [5]. Further antibodies have recently been recognised [6]. The gold standard for diagnosis of MN is a kidney biopsy, although the detection of antibodies against PLA2R is becoming increasingly common and can establish the diagnosis with high degrees of accuracy without the associated risks of a biopsy [7].

The clinical course of PMN is variable and may involve spontaneous remission, relapse or severe NS that progresses to end stage kidney disease (ESKD) [3]. Classically, approximately one third of patients will achieve a spontaneous remission without the need for immunosuppressive treatment [8]. Careful consideration is required to assess the risk of progressive loss of kidney function or complications such as venous thromboembolism to prevent unnecessary use of immunosuppressive agents and their associated side effects. Kidney Diseases Improving Global Outcomes (KDIGO) advocate the use of clinical and laboratory criteria (including serum creatinine, estimated glomerular filtration rate (eGFR), urine protein creatine ratio (uPCR), serum albumin and anti-PLA2R antibodies) to determine this risk. Patients are stratified into low, moderate, high, and very high risk of progressive loss of kidney function [7].

All patients with MN should receive optimal supportive care with use of renin angiotensin system (RAS) blockade to minimise proteinuria. Those at low risk of progressive kidney loss can continue with optimal supportive care, whereas higher risk patients may benefit from immunosuppression. Historically, varying regimens of steroids and cyclophosphamide have been the first line choice for immunosuppression, including the modified Ponticelli (MP) regimen involving a six-month course of alternating monthly cycles of steroids and cyclophosphamide [9]. Recently published KDIGO guidelines (2021) recommend a choice between MP, rituximab or a calcineurin inhibitor (CNI) depending on patient characteristics [7]. Given the toxic potential of cyclophosphamide, MP is now reserved for patients at higher risk of progressive loss of kidney function. A major shift in the management of MN in recent years has been the increasing use of rituximab, due to its superior safety profile. Several trials have demonstrated that it can induce remission with similar efficacy to cyclophosphamide and CNIs, however, the impact on all-cause mortality and progression to ESKD has not been evaluated [10–12].

There have been very few studies assessing large groups of patients with MN over selected time periods. We have undertaken a 20-year retrospective observational study of PMN patients diagnosed at our tertiary renal centre. Our aim was to describe the epidemiology, baseline characteristics, natural disease course, variation in management, and outcomes for this cohort. We believe this provides valuable real-world insight into PMN in a large patient dataset. In addition, we have for the first-time categorised patients as having either low, medium, or high-risk of progression based on KDIGO criteria published in the 2021 glomerulonephritis guidelines and described real-world outcomes.

## Materials and methods

### Patient population

This was a retrospective cohort study conducted on patients diagnosed with PMN at our tertiary renal centre (Salford Royal Hospital, UK), encompassing a catchment population of 1.55 million, between January 2000 and December 2019. The population is largely urban and includes a mixture of affluent areas and those with increased social deprivation.

The Salford Royal kidney biopsy database was screened for patients who were diagnosed with MN between January 2000 and December 2019. Of 238 initially identified, exclusions occurred for the following reasons: diagnosis of secondary MN (39), follow up not at our centre (17), alternative diagnosis (3) or biopsy from a transplant (1). The final study population was 178 patients (Fig 1).

The date of kidney biopsy was used as the study baseline, and all patients were followed until they reached a study endpoint which included (i) commencement of RRT, (ii) death, (iii) end of analysis period (31 December 2020) or (iv) lost to follow up or last documented clinic appointment.

Data on baseline characteristics; laboratory results including creatinine, estimated glomerular filtration rate (eGFR), haemoglobin, albumin, calcium, phosphate, urine protein-creatinine ratio (uPCR); treatment received (including renin-angiotensin system (RAS) blockade and initial immunosuppression); date of initiation of RRT (either transplantation or dialysis); and mortality were gathered from the electronic patient record (EPR). All baseline characteristics and laboratory results were those obtained at the time of or +/- 6 months of biopsy.

A comorbidity of hypertension was defined as a history of hypertension recorded in hospital records, a recorded high blood pressure and/or receiving antihypertensive therapy. A comorbidity of cardiovascular disease included a history of ischaemic heart disease, heart failure, cerebrovascular disease, or peripheral vascular disease.

eGFR values were calculated using the CKD Epidemiology Collaboration (CKD-EPI) formula.

Partial remission was defined as a return to baseline eGFR and uPCR <350g/mol with at least 50% reduction from peak value. Complete remission was defined as return to baseline eGFR and uPCR<25g/mol [13].

### Ethical considerations

The study complies with the declaration of Helsinki and as indicated by the NHS Health Research Authority online tool http://www.hra-decisiontools.org.uk/research this study was not considered research requiring research ethics committee review as it was a retrospective observational study using measurements routinely collected and using fully anonymised and secondary use of data. The need for individual patient consent was waived by the Research and Innovation committee of the Northern Care Alliance NHS Group. The committee granted

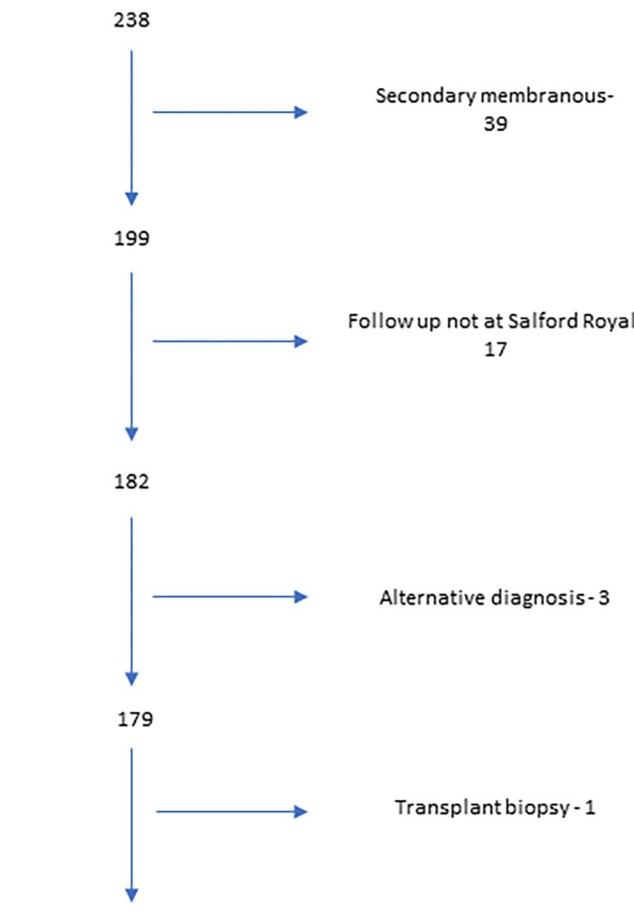

Fig 1. Patient recruitment to the study.

study approval and registered the study (Ref: ID S21HIP41) after approving the methodological protocol as outlined above.

## Statistical analysis

Analysis of baseline characteristics, comorbidities, remission and relapse rates, requirement for RRT, mortality, use of RAS blockade and use and effect of immunosuppression was undertaken in the total cohort. Continuous non-parametric variables are presented as median (interquartile range) and the Mann-Whitney U-test was used to assess between group differences. Categorical data are expressed as number (percentage), and the Chi-square test was used to assess between group differences.

The association of baseline variables with requirement for RRT and mortality was calculated using univariate and multivariate Cox proportional hazard models to determine hazard ratios (HR), 95% confidence intervals (CIs) and statistical significance.

The effect of immunosuppression was determined by comparing those who received immunosuppression and those who did not. The most frequently used specific type of

immunosuppression was Modified Ponticelli, and this group was compared against all other types of immunosuppression combined. Participants were also stratified according to their risk of progression (low, medium, high) based on recently published KDIGO guidelines (2021) [7], and also based on the degree of proteinuria at presentation.

Statistical analysis was performed using IBM SPSS (version 24, University of Manchester).

## Results

We identified a total of 178 patients with primary MN diagnosed between January 2000 and December 2019. The median (IQR) age was 58.3 (44.2–67.1) years, with 63.5% male, 88.8% Caucasian, 10.7% diabetic and 54.5% hypertensive (Table 1). Median serum albumin at presentation was 28g/L (22.3–32.8), median creatinine 90μmol/L (70–122), median eGFR 76.7mls/min/1.73m$^2$ (51.7–90) and median uPCR 664g/mol (393–1007). Anti-PLA2R analysis

**Table 1. Baseline characteristics and outcomes according to immunosuppression vs no immunosuppression.**

| Variable | | Total n = 178 | Immunosuppression n = 94* | No immunosuppression n = 81* | P value |
|---|---|---|---|---|---|
| Age at time of biopsy (years) | | 58.3 (44.2–67.1) | 55.9 (44.1–65.0) | 60.1 (46.0–69.2) | 0.080 |
| Male, *n* (%) | | 113 (63.5) | 67 (71.3) | 44 (54.3) | **0.020** |
| Caucasian, n (%) | | 158 (88.8) | 81 (86.2) | 74 (91.4) | 0.282 |
| Diabetes *n (%)* | | 19 (10.7) | 9 (9.6) | 10 (12.3) | 0.557 |
| Hypertension, *n* (%) | | 97 (54.5) | 46 (48.9) | 50 (61.7) | 0.090 |
| Cardiovascular disease, *n* (%) | | 28 (15.7) | 16 (17.0) | 12 (14.8) | 0.691 |
| SBP, mmHg | | 135 (122–148) | 138 (125.5–150.0) | 130 (120–143) | 0.090 |
| DBP, mmHg | | 79 (70–85) | 80 (70–90) | 75 (70–82) | **0.023** |
| Haemoglobin (g/L) | | 130 (119–143) | 131 (120–144) | 130 (115–153) | 0.650 |
| Albumin (g/L) | | 28 (22.3–32.8) | 26 (21–28) | 31 (25–40) | **<0.001** |
| Corrected Calcium (mmol/L) | | 2.38 (2.28–2.48) | 2.39 (2.31–2.48) | 2.36 (2.27–2.48) | 0.324 |
| Phosphate (mmol/L) | | 1.21 (1.04–1.35) | 1.20 (1.03–1.39) | 1.20 (0.81–1.31) | 0.178 |
| anti-PLA2R positive, *n* (% of those with result available) | | 34 (59.6) | 25 (75.8) | 9 (37.5) | **0.004** |
| anti-PLA2R (U/ml) | | 122 (25.6–190) | 129 (37–274) | 31 (8.5–134.5) | **0.037** |
| eGFR (mls/min/1.73 m$^2$) | | 76.7 (51.7–90) | 78.4 (54.6–90.0) | 72.0 (50.4–90.0) | 0.518 |
| Creatinine (μmol/L) | | 89.5 (70–122) | 86 (71–123) | 92 (70–128.8) | 0.939 |
| uPCR (g/mol) | | 664 (392.5–1006.5) | 829 (542–1151) | 510.5 (311.8–775) | **<0.001** |
| Remission, *n* (%) | Spontaneous | 60 (33.7) | - | 60 (75.9) | - |
| | After treatment | 74 (41.6) | 74 (78.7) | - | - |
| | Total | 134 (75.3) | 74 (78.7) | 60 (75.9) | 0.691 |
| Relapse, *n* (% of those who went into remission) | | 57 (42.5) | 38 (52.1) | 19 (31.7) | **0.008** |
| Received ACEi/ARB, *n (%)* | | 167 (93.8) | 86 (91.5) | 80 (98.8) | **0.030** |
| Progressed to RRT, *n* (%) | | 18 (10.1) | 14 (15.1) | 4 (5.0) | 0.060 |
| Death, *n* (%) | | 53 (29.8) | 25 (26.6) | 28 (34.6) | 0.252 |
| Follow up (months) | | 59.5 (28.8–101.3) | 45 (24.5–96.5) | 68.5 (34–112) | 0.130 |

Continuous variables presented as median (interquartile range), p-value by Mann-Whitney U test. Categorical values presented as number (percentage), p-value by Chi-squared test.

ACEi, angiotensin converting enzyme inhibitor; ARB, angiotensin receptor blocker; DBP, diastolic blood pressure; eGFR, estimated glomerular filtration rate; MN, membranous nephropathy; anti-PLA2R, anti-phospholipase 2A receptor; RRT, renal replacement therapy; SBP, systolic blood pressure; uPCR, urine protein-creatinine ratio.

*Use of immunosuppression data not available for 3 patients

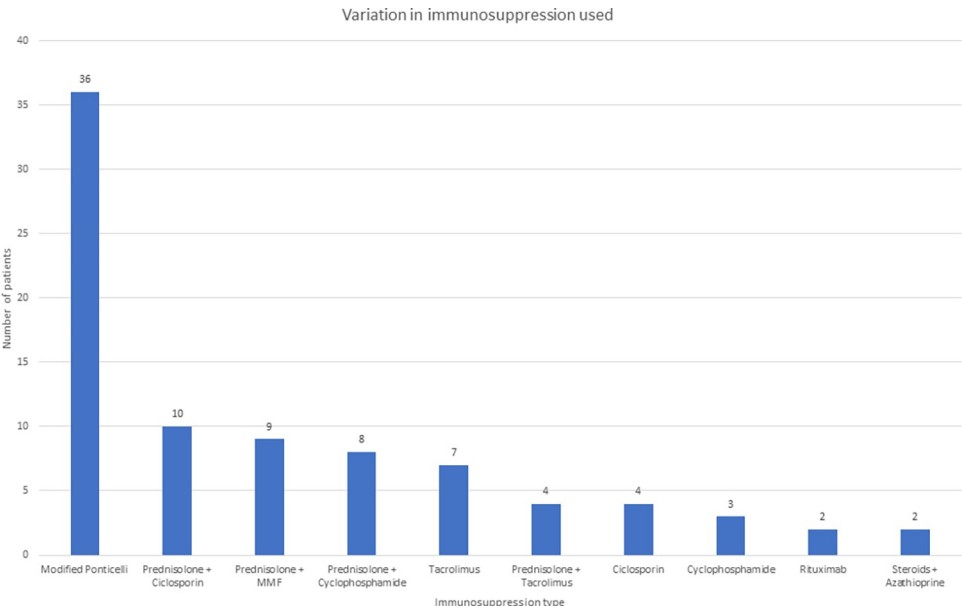

**Fig 2. Initial immunosuppression used.**

was not available during the earlier years of the study period and consequently anti-PLA2R results were only available in 57 patients. Of those tested, 59.6% were positive. Spontaneous remission (either complete or partial) occurred in 60 (33.7%) patients, and of these 19 (31.7%) subsequently relapsed. Remission after immunosuppressive treatment was seen in 74 (41.6%) patients, and of these 38 (51.4%) subsequently relapsed. RAS blockade was used in 93.8% of the cohort. Progression to RRT was only seen in 18 (10.1%) patients and 53 (29.8%) people died during a median follow-up duration of 59.5 months (28.8–101.3).

Immunosuppression was administered to 94 (52.8%) patients, whilst it was not used in 81 patients (45.4%), with data not available for 3 patients. When comparing those who received immunosuppression with those who did not, there were significant differences in gender (71.3% vs 54.3% male, p = 0.02), albumin (26g/L vs 31g/L, p<0.001), uPCR (829g/mol vs 510.5g/mol, p<0.001), relapse rates (51.2% vs 31.7%, p = 0.008), RAS blockade (91.5% vs 98.8%, p = 0.03) and diastolic blood pressure (80mmHg vs 75mmHg, p = 0.023) respectively. Those who received immunosuppression were more likely to require RRT (15.1% vs 5.0%, p = 0.06) and mortality was lower (26.6% vs 34.6%, p = 0.25), although neither of these reached statistical significance. The different types of immunosuppression used, and their frequency are presented in Fig 2.

### Univariate and multivariate Cox regression assessing factors associated with all-cause mortality and progression to RRT

To determine baseline characteristics or laboratory values which were associated with all-cause mortality or need for RRT we performed Cox regression analysis to generate hazard ratios with 95% confidence intervals and p-values (Table 2). Factors associated with all-cause mortality in the univariate models included increasing age (HR 1.03, p = 0.003), diabetes mellitus (HR 1.52, p = 0.017), previous history of cardiovascular disease (HR 2.59, p = 0.002), haemoglobin (HR 0.97, p<0.001), eGFR (HR 0.97, p<0.001) and RAS blockade (HR 0.34, p = 0.043). In the multivariate model, cardiovascular disease (HR 1.96 [1.03–3.71], p = 0.04), eGFR (HR 0.98 [0.96–0.99], p = 0.001) and RAS blockade (HR 0.28 [0.09–0.81], p = 0.019) retained statistical significance.

**Table 2. Association between baseline variables and all-cause mortality and progression to RRT.**

| | All-cause mortality | | | | Progression to RRT | | | |
| --- | --- | --- | --- | --- | --- | --- | --- | --- |
| | Univariate model | | Multivariate model | | Univariate model | | Multivariate model | |
| | Hazard ratio (95% CI) | P-value | Hazard ratio (95% CI) | P-value | Hazard ratio (95% CI) | P-value | Hazard ratio (95% CI) | P-value |
| Age at time of biopsy (years) | 1.03 (1.01–1.06) | **0.003** | 1.02 (0.98–1.04) | 0.092 | 0.99 (0.96–1.02) | 0.407 | - | - |
| Male | 0.98 (0.56–1.72) | 0.949 | - | - | 4.63 (1.06–20.1) | **0.041** | 2.69 (0.59–12.2) | 0.202 |
| Caucasian | 2.63 (0.64–10.8) | 0.181 | - | - | 0.60 (0.17–2.06) | 0.413 | - | - |
| Diabetes | 1.52 (1.08–2.15) | **0.017** | 1.22 (0.84–1.77) | 0.289 | 1.11 (0.53–2.31) | 0.789 | - | - |
| Hypertension | 1.04 (0.59–1.81) | 0.898 | - | - | 13.0 (1.74–98.1) | **0.013** | 4.74 (0.56–39.9) | 0.152 |
| CVD | 2.59 (1.42–4.73) | **0.002** | 1.96 (1.03–3.71) | **0.04** | 0.33 (0.04–2.49) | 0.283 | - | - |
| SBP at biopsy, mmHg | 0.99 (0.98–1.01) | 0.403 | - | | 1.04 (1.02–1.06) | **<0.001** | 1.02 (0.99–1.04) | 0.147 |
| DBP at biopsy, mmHg | 1.00 (0.99–1.00) | 0.424 | - | - | 1.00 (1.00–1.01) | 0.762 | - | - |
| Haemoglobin (g/L) | 0.97 (0.96–0.98) | **<0.001** | 0.99 (0.97–1.01) | 0.197 | 0.96 (0.93–0.98) | **0.001** | 0.99 (0.96–1.02) | 0.665 |
| Albumin (g/L) | 1.01 (0.97–1.05) | 0.767 | - | - | 1.01 (0.94–1.08) | 0.835 | - | - |
| eGFR (mls/min/1.73 m$^2$) | 0.97 (0.96–0.98) | **<0.001** | 0.98 (0.96–0.99) | **0.001** | 0.95 (0.93–0.97) | **<0.001** | 0.96 (0.94–0.99) | **0.005** |
| uPCR (g/mol) | 1.001 (1.00–1.001) | 0.060 | - | - | 1.00 (1.00–1.00) | 0.133 | - | - |
| Received ACEi/ ARB | 0.34 (0.12–0.97) | **0.043** | 0.28 (0.09–0.81) | **0.019** | 0.60 (0.08–4.54) | 0.618 | - | - |
| Received immunosuppression | 0.91 (0.53–1.56) | 0.727 | - | - | 3.34 (1.10–10.2) | **0.034** | 3.99 (1.11–14.4) | **0.034** |

ACEi/ARB, angiotensin converting enzyme inhibitor/ angiotensin receptor blocker; CI, confidence interval; CVD, cardiovascular disease; DBP, diastolic blood pressure; eGFR, estimated glomerular filtration rate; RRT, renal replacement therapy; SBP systolic blood pressure; uPCR, urine protein-creatinine ratio

Factors associated with need for RRT included male gender (HR 4.63, p = 0.041), hypertension (HR 13.0, p = 0.013), systolic blood pressure (HR 1.04, p<0.001), haemoglobin (HR 0.96, p = 0.001), eGFR (HR 0.95, p<0.001) and use of immunosuppression (HR 3.34, p value = 0.034). In the multivariate model only eGFR (HR 0.96 [0.94–0.99], p = 0.005) and use of immunosuppression (HR 3.99 [1.11–14.4], p = 0.034) retained their significance. It is worth noting that use of immunosuppression being related to increased risk of progression to RRT is likely due to confounding factors such as more severe form of disease at presentation influencing the decision to use immunosuppression.

## Modified Ponticelli vs other immunosuppression

Of those who received immunosuppression, the largest category was MP (n = 36). The baseline characteristics, laboratory values and outcomes for those who received MP and those who received other forms of immunosuppression can be seen in Table 3. When comparing these 2 groups, there were fewer relapses (38.7% vs 66.7%, p = 0.02) in those who received MP, increased remission rates (88.9% vs 71.9%, p = 0.053), reduced progression to RRT (2.8% vs 22.8%, p = 0.02), lower mortality (11.1% vs 36.2%, p = 0.007), less co-existing hypertension (33.3% vs 58.6%, p = 0.017), and higher calcium levels (2.43mmol/L [2.36–2.59] vs 2.37mmol/ L [2.22–2.48], p = 0.015).

There were 58 patients who received non-MP immunosuppression. Fig 2 depicts the full range of different immunosuppression types administered which includes prednisolone in combination with CNIs, MMF, cyclophosphamide and azathioprine; along with CNIs alone, cyclophosphamide alone and rituximab alone. The variety of immunosuppression use reflects the long study period in which immunosuppression use was less standardised prior to the introduction of KDIGO guidelines in 2012.

**Table 3. Baseline characteristics and outcomes according to whether a patient received the modified Ponticelli regimen vs other forms of immunosuppression.**

| Variable | Modified Ponticelli, n = 36 | Other, n = 58 | P value |
|---|---|---|---|
| Age at time of biopsy (years) | 54.6 (39.5–65.1) | 61.2 (48.2–65.3) | 0.094 |
| Male, *n* (%) | 26 (72.2) | 41 (70.7) | 0.873 |
| Caucasian, *n* (%) | 34 (94.4) | 47 (81.0) | 0.067 |
| Diabetes *n* (%) | 2 (5.6) | 7 (12.1) | 0.297 |
| Hypertension, *n* (%) | 12 (33.3) | 34 (58.6) | **0.017** |
| Cardiovascular disease, *n* (%) | 9 (25.0) | 7 (12.1) | 0.105 |
| Systolic BP, mmHg | 132.5 (126.3–151.3) | 140 (120–150) | 0.956 |
| Diastolic BP, mmHg | 79.5 (70–89.5) | 80.0 (70–90) | 0.622 |
| Haemoglobin (g/L) | 131 (120–143.3) | 129.5 (119.5–144) | 0.828 |
| Albumin (g/L) | 26.5 (22–28) | 26 (20–28.8) | 0.934 |
| Corrected calcium (mmol/L) | 2.43 (2.36–2.49) | 2.37 (2.22–2.48) | **0.015** |
| Phosphate (mmol/L) | 1.18 (1.02–1.39) | 1.26 (1.09–1.38) | 0.468 |
| anti-PLA2R positive, *n* (% of those with result available) | 20 (76.9) | 5 (71.4) | 0.763 |
| anti-PLA2R (U/ml) | 158 (41.5–366) | 52 (29.5–136.5) | 0.129 |
| eGFR (mls/min/1.73 m$^2$) | 81.6 (59.4–90) | 77.2 (49.2–90.0) | 0.256 |
| Creatinine (μmol/L) | 80.5 (71–117.8) | 91 (70.8–146.8) | 0.217 |
| uPCR (g/mol) | 826.5 (589–1149) | 793 (429.8–1452) | 0.562 |
| Remission, *n* (%) | 32 (88.9) | 41 (71.9) | 0.053 |
| Relapse, *n* (% of those who went into remission) | 12 (38.7) | 26 (66.7) | **0.020** |
| Received ACEi/ARB, *n (%)* | 35 (97.2) | 51 (87.9) | 0.117 |
| Progressed to RRT *n* (%) | 1 (2.8) | 13 (22.8) | **0.021** |
| Death, *n* (%) | 4 (11.1) | 21 (36.2) | **0.007** |
| Follow up (months) | 47 (24.5–81.8) | 45 (24.3–126) | 0.276 |

Continuous variables presented as median (interquartile range), p-value by Mann-Whitney U test. Categorical values presented as number (percentage), p-value by Chi-squared test.

ACEi, angiotensin converting enzyme inhibitor; ARB, angiotensin receptor blocker; DBP, diastolic blood pressure; eGFR, estimated glomerular filtration rate; MN, membranous nephropathy; anti-PLA2R, anti-phospholipase 2A receptor; RRT, renal replacement therapy; SBP, systolic blood pressure; uPCR, urine protein-creatinine ratio.

## Remission rates (partial, complete or no remission)

Within the total cohort, 39 (22.5%) patients did not achieve remission, 65 (37.6%) achieved partial remission and 69 (39.9%) achieved complete remission (Table 4). Spontaneous remission was seen in 60 (33.7%), whilst remission after treatment with immunosuppression was seen in 74 (41.6%). Remission data was not available for 5 patients due to lack of available data. Baseline characteristics were compared between those who did not achieve remission and those who achieved complete remission. There were several significant differences between these groups (Table 4) including mortality (67% vs 7%, p<0.001) and progression to RRT (28% vs 1%, p<0.001). This is also shown in the Kaplan-Meier curves in Fig 3 for overall survival (A) and freedom from RRT (B). There were further differences between the 2 groups with respect to age (65 years [44–73] vs 56 years [45–64], p = 0.046), eGFR (47mls/min/1.73m$^2$ [21–84] vs 89mls/min/1.73m$^2$ [76–90], p<0.001), creatinine (134μmol/L [82–284] vs 76μmol/L [62–92], p<0.001), uPCR (976g/mol [537–1267] vs 564g/mol [325–975], p = 0.002), RAS blockade (85% vs 96%, p = 0.049), diabetes (23% vs 3%, p = 0.001), hypertension (62% vs 40%, p = 0.030), haemoglobin (122g/L [111–135] vs 134g/L [125–147], p = 0.002), and follow up

**Table 4. Baseline characteristics and outcomes by remission (no, partial, or complete).**

| Variable | No remission n = 39* | Partial remission n = 65* | Complete remission n = 69* | P value (For no vs complete remission) |
|---|---|---|---|---|
| Age at time of biopsy (years) | 65 (44–73) | 59 (45–68) | 56 (45–64) | **0.046** |
| Male, *n* (%) | 26 (67) | 45 (69) | 39 (57) | 0.301 |
| Caucasian, *n* (%) | 34 (87) | 60 (92) | 59 (86) | 0.809 |
| Diabetes *n* (%) | 9 (23) | 8 (12) | 2 (3) | **0.001** |
| Hypertension, *n* (%) | 24 (62) | 43 (66) | 27 (40) | **0.030** |
| Cardiovascular disease, *n* (%) | 9 (23) | 10 (15) | 9 (13) | 0.179 |
| Systolic BP, mmHg | 133 (116–156) | 140 (130–150) | 130 (120–145) | 0.885 |
| Diastolic BP, mmHg | 78 (69–83) | 80 (70–90) | 77 (70–84) | 0.742 |
| Haemoglobin (g/L) | 122 (111–135) | 130 (114–145) | 134 (125–147) | **0.002** |
| Albumin (g/L) | 28 (21–32) | 27 (22–33) | 27 (23–33) | 0.425 |
| Corrected Calcium (mmol/L) | 2.38 (2.20–2.52) | 2.38 (2.29–2.46) | 2.37 (2.30–2.46) | 0.966 |
| Phosphate (mmol/L) | 1.18 (1.09–1.42) | 1.23 (1.05–1.42) | 1.18 (1.03–1.32) | 0.322 |
| anti-PLA2R positive, *n* (%) | 4 (10) | 16 (25) | 13 (19) | 0.447 |
| anti-PLA2R (U/ml) | 98 (32–150) | 122 (31–152) | 61 (9–294) | 0.920 |
| eGFR (mls/min/1.73 m$^2$) | 47 (21–84) | 63 (51–90) | 89 (76–90) | <**0.001** |
| Creatinine (μmol/L) | 134 (82–284) | 96 (77–129) | 76 (62–92) | <**0.001** |
| uPCR (g/mol) | 976 (537–1267) | 611 (399–812) | 564 (325–975) | **0.002** |
| Received ACEi/ARB, *n (%)* | 33 (85) | 65 (100) | 65 (96) | **0.049** |
| Received immunosuppression, *n* (%) | 20 (51) | 39 (60) | 34 (49) | 0.898 |
| Progression to RRT, *n* (%) | 11 (28) | 5 (8) | 1 (1) | <**0.001** |
| Death, *n* (%) | 26 (67) | 22 (34) | 5 (7) | <**0.001** |
| Follow up (months) | 29 (9–45) | 81 (28–126) | 83 (46–129) | <**0.001** |

*Remission data not available for 5 patients

Continuous variables presented as median (interquartile range), p-value by Mann-Whitney U test. Categorical values presented as number (percentage), p-value by Chi-squared test.

ACEi, angiotensin converting enzyme inhibitor; ARB, angiotensin receptor blocker; DBP, diastolic blood pressure; eGFR, estimated glomerular filtration rate; MN, membranous nephropathy; anti-PLA2R, anti-phospholipase 2A receptor; RRT, renal replacement therapy; SBP, systolic blood pressure; uPCR, urine protein-creatinine ratio.

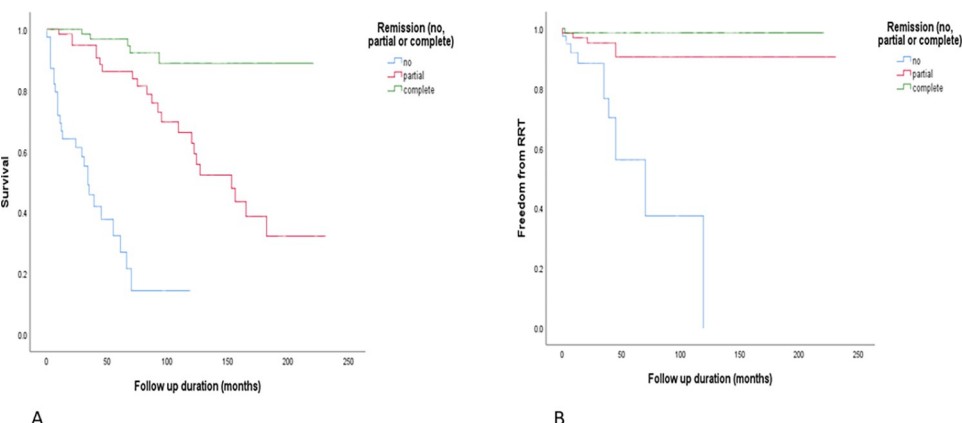

**Fig 3.** Overall survival (A) and freedom from RRT (B) in patients with no remission, partial remission, and complete remission. P-values <0.001 and <0.001 respectively (Chi-squared).

duration (29 months [9–45] vs 83 months [46–129], p<0.001) for no remission and complete remission, respectively.

## Degree of proteinuria at biopsy

Patients were grouped according to degree of proteinuria at time of biopsy (mild, uPCR <350g/mol; moderate, uPCR 350–800 g/mol; severe uPCR >800g/mol). The baseline characteristics, laboratory values and outcomes for these groups can be seen in S1 Table. There were significant differences in the following variables when the severe proteinuric group was compared to the mild group: more with diabetes (18% vs 0%, p = 0.008); lower albumin (26g/L [20–28] vs 34 g/L [28–39]); reduced rates of remission (65% vs 86%, p = 0.015) and increased rates of immunosuppression use (73% vs 34%, p<0.001). There were no statistically significant differences in outcomes between the severe proteinuric and mild proteinuric group (progression to RRT in 14% vs 14%, p = 0.790, and mortality in 33% vs 26%, p = 0.508).

## Risk of progression (low/ medium/ high)

Patients were categorised as having either low, medium or high risk of progression based on KDIGO criteria published in the 2021 glomerulonephritis guidelines [7]. The criteria include eGFR, albumin, proteinuria and other factors indicating high-risk of progression such as anti-PLA2R value. We identified 83 patients at low-risk of progression, 15 patients with medium-risk of progression and 76 patients at high-risk of progression (Table 5).

## Timing of biopsy

Finally, we also split the cohort according to timing of biopsy (2000–2011 and 2012–2019). The results can be seen in S2 Table.

The high-risk group was compared with the low-risk group. The high-risk group was older (median age at diagnosis 61.2 [48.9–68.1] years vs 54.2 [40.8–66.0] years, p = 0.029), more likely to be hypertensive (68.4% vs 48.2%, p = 0.003), had a lower haemoglobin (124.5g/L [108–137] vs 134g/L [122–153], p<0.001), lower eGFR (49ml/min/1.73m$^2$ [26.5–58.5] vs 90ml/min/1.73m$^2$ [65–90], p<0.001), less likely to develop remission (56% vs 97.6%, p<0.001), more likely to progress to RRT (8.1% vs 3.7%, p = 0.012) and more likely to die (52.6% vs 10.8%, p<0.001). Treatment approaches were similar across the 2 groups with 39 (51.3%) receiving immunosuppression in the high-risk group, and 46 (55.4%) receiving immunosuppression in the low-risk group. Of those who received immunosuppression in the high-risk group, 12 (30.1%) received MP whereas 20 (43.5%) received MP in the low-risk group. The high-risk group also had a shorter duration of follow up (42 months [19.5–118] vs 74 months [43–99], p = 0.009) likely reflecting the increased rate of outcomes observed in this group. Kaplan-Meier curves for overall survival and freedom from RRT for those with low, medium, and high-risk of progression can be seen in Fig 4(A) and 4(B).

## Discussion

This is a large retrospective epidemiological study for PMN which provides important real-world information about the natural history of this condition. This is also, to the best of our knowledge, the first study to categorise patients into low, medium, and high-risk of progression as per the 2021 KDIGO guidelines. Key findings from the study are that i) progression to RRT was only seen in a minority (10.1%), in contrast to the previous literature, ii) most patients presented with low serum albumin and nephrotic range proteinuria (median serum albumin 26g/L and median uPCR 664g/mol); and iii) the majority went into remission

**Table 5. Baseline characteristics and outcomes according to risk of progression as per KDIGO (low/ medium/ high).**

| Variable | | Low risk, n = 83* | Medium risk, n = 15* | High risk, n = 76* | P value (For low-risk vs high-risk) |
|---|---|---|---|---|---|
| Age at time of biopsy (years) | | 54.2 (40.8–66.0) | 55.0 (44.2–63.3) | 61.2 (48.9–68.1) | **0.029** |
| Male, n (%) | | 51 (61.4) | 7 (46.7) | 52 (68.4) | 0.251 |
| Caucasian, *n* (%) | | 72 (86.7) | 12 (80) | 70 (92.1) | 0.319 |
| Diabetes, *n* (%) | | 7 (8.4) | 3 (20) | 9 (11.8) | 0.394 |
| Hypertension [b], *n* (%) | | 40 (48.2) | 4 (26.7) | 52 (68.4) | **0.003** |
| Cardiovascular disease, *n* (%) | | 10 (12) | 1 (6.7) | 17 (22.4) | 0.122 |
| Systolic BP, mmHg | | 134 (125–148) | 132 (118–141) | 137.5 (121–150) | 0.642 |
| Diastolic BP, mmHg | | 80 (70–95) | 78 (66–85) | 77.5 (68.8–83) | 0.237 |
| Haemoglobin (g/L) | | 134 (126–147) | 134 (122–153) | 124.5 (108–137) | **<0.001** |
| Albumin (g/L) | | 27 (22–32) | 28 (24–21) | 28 (23–34) | 0.416 |
| Corrected Calcium (mmol/L) | | 2.39 (2.32–2.48) | 2.39 (2.32–2.51) | 2.36 (2.21–2.47) | 0.064 |
| Phosphate (mmol/L) | | 1.20 (1.00–1.32) | 1.18 (0.96–1.39) | 1.22 (1.06–1.47) | **0.029** |
| anti-PLA2R positive, *n* (% of those with result available) | | 22 (61.1) | 6 (100) | 6 (40) | **0.039** |
| anti-PLA2R (U/ml) | | 122.5 (33–260) | 55 (27–136) | 131 (15–179) | 0.764 |
| eGFR (mls/min/1.73 m$^2$) | | 90 (76.7–90) | 90 (64.5–90) | 49.1 (26.5–58.5) | **<0.001** |
| Creatinine (μmol/L) | | 74 (62.5–90.3) | 80 (64–92) | 132.5 (101.5–202) | **<0.001** |
| Remission, *n* (%) | Spontaneous | 36 (43.9) | 4 (30.8) | 19 (25.3) | **0.048** |
| | After treatment | 44 (53.7) | 6 (46.2) | 23 (30.7) | **0.014** |
| | Total | 80 (97.6) | 10 (76.9) | 42 (56) | **<0.001** |
| Relapse, *n* (% of those who went into remission) | | 35 (44.9) | 2 (22.2) | 20 (46.5) | 0.393 |
| Received ACEi/ARB, *n (%)* | | 80 (96.4) | 15 (100) | 70 (92.1) | 0.305 |
| Received immunosuppression, *n* (%) | | 46 (55.4) | 9 (64.3) | 39 (51.3) | 0.605 |
| Progression to RRT, *n* (%) | | 3 (3.7) | 0 (0) | 14 (8.1) | **0.012** |
| Death, *n* (%) | | 9 (10.8) | 4 (26.7) | 40 (52.6) | **<0.001** |
| Follow up (months) | | 74 (43–99) | 41 (19–92) | 42 (19.5–118) | **0.009** |

*Data not available for 4 patients

Continuous variables presented as median (interquartile range), p-value by Mann-Whitney U test. Categorical values presented as number (percentage), p-value by Chi-squared test.

ACEi, angiotensin converting enzyme inhibitor; ARB, angiotensin receptor blocker; DBP, diastolic blood pressure; eGFR, estimated glomerular filtration rate; MN, membranous nephropathy; anti-PLA2R, anti-phospholipase 2A receptor; RRT, renal replacement therapy; SBP, systolic blood pressure; uPCR, urine protein-creatinine ratio.

(spontaneous 33.7%, and after treatment 41.6%)—but of these a significant number relapsed (42.5%). Patients who received the modified Ponticelli immunosuppressive regime had significantly better outcomes than those who received other forms of immunosuppression. The value in risk stratifying patients according to their risk of progression was confirmed; those at high risk had far worse outcomes compared to those at low risk.

There are few long-term prospective or retrospective studies of PMN reported in the literature. Rozenberg et al. [14] performed a retrospective study of PMN in 2018 which analysed 16 patients considered as being at high-risk of progression and treated with immunosuppression, and 21 patients considered as low risk who received supportive care. Remission rates were 68.7% in the high-risk group (in comparison to 56% in our cohort), and 90.4% in the low-risk group (in comparison to 97.6% in our cohort). Progression to RRT was seen in 25% of the high-risk group (compared to only 8.1% in our cohort). A group from New Zealand assessed the outcomes of MN (amongst other glomerulonephritidies) in a cohort who were initially

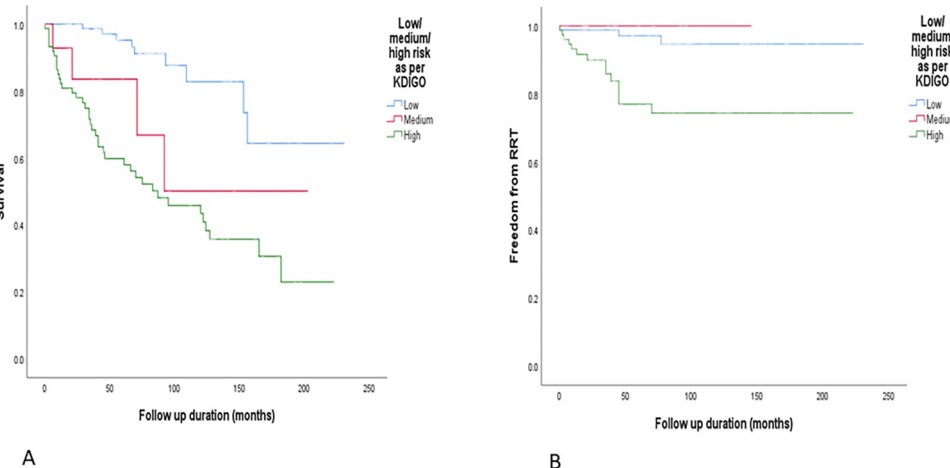

**Fig 4.** Overall survival (A) and freedom from RRT (B) in patients categorised as low, medium, and high-risk of progression. P values <0.001 and 0.001 respectively (Chi-squared).

enrolled into the New Zealand Glomerulonephritis Study between 1972 and 1983 [15]. The cohort included 87 patients with MN, and of these 20% had developed ESKD at 10 years.

Hamano et al. identified 67 patients with MN as part of the Chronic Kidney Disease Japan Cohort (CKD-JAC) study- a multicentre prospective observational cohort study which enrolled patients between April 2007 and December 2008. They found an incidence rate of progression to ESKD amongst MN patients to be 35.3/1000 person years [16].

A retrospective study in London identified 148 MN patients between 1995 and 2015 [17]. It found that those who presented with a higher creatinine were more likely to develop RRT, and surprisingly, those who progressed to RRT were more likely to have a higher serum albumin at presentation than those who did not. Black patients (comprising 24% of their cohort) had a worse outcome and poorer response to treatment. Within this cohort 36% were White, whereas 88.8% were of Caucasian ethnicity in our cohort. This suggests that there may be outcome differences based on ethnicity that requires further evaluation in multi-ethnic cohorts.

Other retrospective studies have been undertaken in China [18], Korea [19], Japan [20, 21], Australia and New Zealand [22], USA [23] and the UK [24]. In an elderly cohort from Korea [19], immunosuppression use was found to increase the likelihood of adverse renal outcomes and increased rates of infection, whereas RAS blockade reduced both, suggesting that use of immunosuppression should be carefully considered in the elderly. A UK study including patients diagnosed between 1980 and 2010 identified 128 patients with PMN [24]. This study reported similar remission rates to our study, however of the 21.9% that did not achieve remission, 75% of these patients went on to develop ESKD in comparison to only 28% in our study. This is likely due to the earlier time period of the study in an era when RAS blockade was not universal and immunosuppression use was more limited. The MP regimen was established into clinical practice in the late 1990s after evidence emerged for the beneficial effect of alternating monthly cycles of steroids and either chlorambucil or cyclophosphamide [25]. Our study suggests that changes in treatment, in particular with standardisation of practice with immunosuppression use, has improved outcomes for patients with PMN in a real-world setting.

We observed a wide variety of types of immunosuppression used, with MP (38 patients) being the largest category. The reason for the disparity in immunosuppression types can be explained by the long time-period of the study (20 years) with KDIGO guidelines published in

2012 likely contributing to a greater degree of standardisation in clinical practice thereafter. It is interesting to note the superior outcomes achieved with MP compared to other immunosuppression when examined in the immunosuppressed sub-group. It is also important to bear in mind that not all patients who start MP treatment complete the 6-month course. These results should be interpreted with caution as the comparison contains many different types of immunosuppression within the non-MP cohort. A better comparison would be MP vs another specific type of immunosuppression treatment such as calcineurin inhibitors or rituximab, but we were unable to perform this analysis due to the low numbers attributed to these individual immunosuppression treatment types within our cohort. Furthermore, the MP results may be confounded by intention to treat bias- it is possible that we did not give MP to those who were felt to be more susceptible to its risks, for example the frail elderly, who more likely to have worse outcomes regardless of the immunosuppression type used.

Our centre first began testing anti-PLA2R antibody levels in 2012 and as such our anti-PLA2R data is far from complete. A future study would aim to collect more comprehensive anti-PLA2R data and look at factors such as whether antibody titres correlate with likelihood of remission, severity of disease and whether levels can predict risk of relapse, as reported in previous studies [26, 27].

One area that our study did not assess was thrombo-embolic risk with IMN. Zou et al. assessed the incidence of venous thromboembolism (VTE) and arterial thromboembolism in a cohort of 766 patients in China [28]. The highest risk was seen within the first 6 months after diagnosis, reaching 8.0% for arterial thromboembolism and 7.2% for venous thromboembolism at 5 years post diagnosis. Hypoalbuminaemia was found to be the dominant independent risk factor for VTE, with nephrotic syndrome present in 81.5% of patients who had a VTE.

Finally, the study time period was largely before the sodium-glucose co-transport-2 inhibitors (SGLT2i) era. Given the growing evidence for the use of SGLT2i in proteinuric kidney disease regardless of cause [29], we suggest that they should be considered for overall nephroprotection in patients with MN.

## Conclusion

This large real-world study has identified several key points. Only 10.1% of our cohort progressed to ESKD during the follow up period, suggesting that outcomes for PMN are improving. Those who received MP had better outcomes (less relapse, less progression to RRT and reduced mortality) compared to those treated with other immunosuppression. We were not able to assess the use of rituximab due to the time-period of the study resulting in only 2 patients receiving it. A high rate of remission was observed (75.3%), although a significant number of these (42.9%) subsequently relapsed. Failure to achieve remission was associated with worse outcomes. Finally, when comparing patients by risk status according to the KDIGO guidelines those classified as at high-risk were more likely to progress to RRT and had a higher mortality, highlighting the clinical utility of this classification.

## Supporting information

**S1 Table. Baseline characteristics and outcomes according to degree of proteinuria at biopsy.**
(DOCX)

**S2 Table. Baseline characteristics and outcomes according to timing of biopsy—2000–2011 vs 2012–2019.**
(DOCX)

**S1 Dataset.**
(XLSX)

## Author Contributions

**Conceptualization:** Philip A. Kalra, Smeeta Sinha.

**Data curation:** Joshua Storrar, Tarra Gill-Taylor.

**Formal analysis:** Tarra Gill-Taylor, Rajkumar Chinnadurai.

**Funding acquisition:** Philip A. Kalra.

**Methodology:** Joshua Storrar, Philip A. Kalra, Smeeta Sinha.

**Supervision:** Philip A. Kalra, Smeeta Sinha.

**Writing – original draft:** Joshua Storrar, Tarra Gill-Taylor.

**Writing – review & editing:** Joshua Storrar, Rajkumar Chinnadurai, Constantina Chrysochou, Dimitrios Poulikakos, Francesco Rainone, James Ritchie, Elizabeth Lamerton, Philip A. Kalra, Smeeta Sinha.

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
