## [Decision Letter · Decision Letter 0]

12 Aug 2022

PONE-D-22-19331A low rate of end-stage kidney disease in membranous nephropathy: a single centre study over 2 decadesPLOS ONE

Dear Dr. Storrar,

Thank you for submitting your manuscript to PLOS ONE. After careful consideration, we feel that it has merit but does not fully meet PLOS ONE’s publication criteria as it currently stands. Therefore, we invite you to submit a revised version of the manuscript that addresses the points raised during the review process.

We look forward to receiving your revised manuscript.

Kind regards,

Donovan Anthony McGrowder, PhD., MA., MSc

Academic Editor

PLOS ONE

Journal Requirements: 

"I have read the journal's policy and the authors of this manuscript have the following competing interests:

S Sinha has received grants from Johnson and Johnson and AstraZeneca; speaker and lecture fees from AstraZeneca, Napp, Bayer, Sanofi-Genzyme, Novartis, and Vifor Pharma; is on advisory boards for Novartis, AstraZeneca, Bayer, and Travere; and has a clinical consultancy role with Sanifit. 

PA Kalra has received grants from Astellas, Vifor Pharma, BergenBio and Evotec; speaker and lecture fees from AstraZeneca, Napp, Bayer, Novartis, Vifor Pharma, Pharmacosmos, Boehringer Ingelheim; is on advisory boards for AstraZeneca and Vifor Pharma; and has a consultancy role with Bayer, Astella, Otsuka and Unicyte."

Additional Editor Comments:

Dear Dr. Storrar,

Your manuscript “A low rate of end-stage kidney disease in membranous nephropathy: a single centre study over 2 decades” has been assessed by our reviewers. They have raised a number of points which we believe would improve the manuscript and may allow a revised version to be published in PLOS ONE. Their reports, together with any other comments, are below.

If you are able to fully address these points, we would encourage you to submit a revised manuscript to PLOS ONE.

Best regards,

Dr. Donovan McGrowder

Reviewers' comments:

Reviewer's Responses to Questions

**Comments to the Author**

1. Is the manuscript technically sound, and do the data support the conclusions?

Reviewer #1: Yes

Reviewer #2: No

Reviewer #3: Yes

Reviewer #4: Yes

2. Has the statistical analysis been performed appropriately and rigorously? 

Reviewer #1: Yes

Reviewer #2: No

Reviewer #3: Yes

Reviewer #4: Yes

3. Have the authors made all data underlying the findings in their manuscript fully available?

Reviewer #1: Yes

Reviewer #2: Yes

Reviewer #3: Yes

Reviewer #4: Yes

4. Is the manuscript presented in an intelligible fashion and written in standard English?

Reviewer #1: Yes

Reviewer #2: Yes

Reviewer #3: Yes

Reviewer #4: Yes

5. Review Comments to the Author

Reviewer #1: This is an excellent report summarizing the diagnosis, treatment and prognosis of primary membranous nephropathy patients in the real world. As described in the text, a more accurate diagnosis of primary membranous nephropathy can be made recently by adding serodiagnosis such as PLA2R antibody. Therefore, the number of the patients who progress to ESKD is smaller than previously thought, and supportive care is favorable in milder cases than using immunosuppressive therapy which is likely to cause adverse events. Given the growing evidence for SGLT2 inhibitors as a means of supportive care in CKD, consideration about using SGLT2 inhibitors may be added in the discussion.

Reviewer #2: The authors investigated the association between the clinical factors, including the severity or treatment, and mortality or the incidence of ESKD among 178 patients with primary MN. Although the results were partially meaningful, there were several problems. Although they used the proportion of death or progression to RRT as markers for outcomes of interest, this parameters are affected by the observational period. For example, the follow-up periods of the patients without remission was clearly shorter than those of the patients with partial or complete remission in Table 4. Therefore, the proportion of death or progression to RRT could be underestimated in the patients with former group. I recommend using times per patient-month or patient-year as a marker for outcome. Explanatory variables of Table 2 to 5 should be limited to baseline characteristics. It is strange that they compared the proportion of relapse between patients without remission, those with partial remission and those with complete remission in Table 4.

Although the title was a low rate of ESKD in MN, they tried to identify the risk factors for death or ESKD among patients with MN. At least, they should cited previous reports showing the rate of ESKD in MN. Chembo et al. reported that less than 20% of patients with MN had reached ESKD at 10 years follow up using data obtained from the New Zealand Glomerulonephritis Study [Chembo CL et al. Nephrology (Carlton) 20:899-907,2015]. Recently, Hamano et al. reported that incidence of dialysis among patients with MN was approximately 30/1000 person-years, and the risk was 0.45 times lower than patients with IgA nephropathy using the CKD-JAC Study, a multicenter, prospective observational cohort study in Japan. [Hamano T et al, Nephrol Dial Transplant. 2022 Mar 22:gfac134. Online ahead of print].

To investigate the clinical efficacy of the modified Ponticelli immunosuppressive regime, control group should be patients without immunosuppression.

In the Conclusion, the authors only repeated to present the results.

Reviewer #3: Storrar J et al described retrospective analysis of 178 primary membranous nephropathy (PMN) patients regarding outcomes of remission, renal replacement therapy and all cause mortality. This study enrolled medium number of patients in a single center in UK. I think the real-world data regarding PMN are clinically relevant and the manuscript is worth reading for the nephrologists. I have some comments which need to be addressed.

1. The authors showed that 33.7% of patients reached spontaneous complete or partial remission, and 41.6% of patients reached remission by immunosuppressive treatment. In case of spontaneous remission, how long do you observe the patients without immunosuppression treatment? What is the inducement of the initiation of immunosuppression therapy in the patients with MN?

2. The patients are enrolled from 2000 until 2019. During two decades of study period, therapy for PMN was changed substantially. Could the authors compare the remission rate between early period from 2000 to 2010 and late period from 2011 to 2019?

3. Please show the Kaplan-Mayer analysis for cumulative probabilities of remission against follow-up period.

4. Deaths was reported 53 cases out of 178 patients. Approximately 30% of patients were die. It is very high if the mortality was caused by kidney diseases. Please identify the cause of death.

Reviewer #4: The work is of clinical value in that they employ the recent categorization patients with PMN into low, medium, and high-risk of progression as per the 2021 KDIGO guidelines.

Major concerns,

1) With respect to the ratio of remission or RRT, the ratio is not appropriate, but should be presented as the incidence of RRT with a unit of /patient/yr.

2) To define the partial and complete remission of proteinuria, please put the appropriate references.

3) Please perform the subgroup analysis after blood anti-PLA2R antibody level was started to be assessed routinely even if “our anti-PLA2R data is far from complete”. Or, please specify how incomplete they are. That topic is what not a few nephrologists are so interested in.

Minor concerns,

1) In the Abstract, MN should be spelled out.

2) In the Abstract, specify the mean or median of the observation period.

3) In the Abstract, the referee does not understand “Amongst the whole cohort, those who went into remission did better than those who did not.” Also, without the definition of high- or low-risk, it is difficult to understand “Those classified as high-risk also had worse outcomes than those at low-risk”.

---

## [Author Response · Author response to Decision Letter 0]

22 Sep 2022

In response to your points:

1. The manuscript meets PLOS ONE’s style requirements.

2. We have added the required ‘competing interests’ statement to the cover letter as requested.

Dear reviewers,

Many thanks for all of your comments. Our responses are detailed below.

Reviewer 1

Given the growing evidence for SGLT2 inhibitors as a means of supportive care in CKD, consideration about using SGLT2 inhibitors may be added in the discussion.

This has been commented on in the discussion.

Reviewer 2

Although they used the proportion of death or progression to RRT as markers for outcomes of interest, these parameters are affected by the observational period. For example, the follow-up periods of the patients without remission was clearly shorter than those of the patients with partial or complete remission in Table 4. 

Therefore, the proportion of death or progression to RRT could be underestimated in the patients with former group. I recommend using times per patient-month or patient-year as a marker for outcome. 

The reason for the follow up period being shorter amongst those who do not achieve remission is because the end date for follow up period includes progression to RRT and death. If they have more of these events then it follows that the follow up duration will be less. The proportion who progressed to RRT or died will not be underestimated as these parameters were assessed in all patients. As such, whilst we recognise the utility of per patient-month/ per patient-year, we have not switched to this for our analysis. 

Explanatory variables of Table 2 to 5 should be limited to baseline characteristics. It is strange that they compared the proportion of relapse between patients without remission, those with partial remission and those with complete remission in Table 4.

Variables assessed in table 2 are all baseline variables, consistent with performing Cox-regression analysis. Variables in tables 3-5 include baseline and outcome measures such as such as progression to RRT and death as these outcomes are a key part of the study.

We have removed the comparison of relapse rates in table 4. 

Although the title was a low rate of ESKD in MN, they tried to identify the risk factors for death or ESKD among patients with MN. At least, they should cite previous reports showing the rate of ESKD in MN. 

Chembo et al. reported that less than 20% of patients with MN had reached ESKD at 10 years follow up using data obtained from the New Zealand Glomerulonephritis Study [Chembo CL et al. Nephrology (Carlton) 20:899-907,2015]. 

Recently, Hamano et al. reported that incidence of dialysis among patients with MN was approximately 30/1000 person-years, and the risk was 0.45 times lower than patients with IgA nephropathy using the CKD-JAC Study, a multicenter, prospective observational cohort study in Japan. [Hamano T et al, Nephrol Dial Transplant. 2022 Mar 22:gfac134. Online ahead of print].

Many thanks for highlighting these, we have referenced them in the discussion.

To investigate the clinical efficacy of the modified Ponticelli immunosuppressive regime, the control group should be patients without immunosuppression.

The reason that we used the control group of ‘other immunosuppression’ was because we have already used a control group of ‘no immunosuppression’ in table 1. Ideally we would have liked to compare individual specific types of immunosuppression, but as commented in the paper, this was not possible due to the low numbers in each of the different categories (see figure 2 for a full breakdown of these).

In the Conclusion, the authors only repeated to present the results.

We have amended the conclusion to provide more analysis.

Reviewer 3

1. The authors showed that 33.7% of patients reached spontaneous complete or partial remission, and 41.6% of patients reached remission by immunosuppressive treatment. In case of spontaneous remission, how long do you observe the patients without immunosuppression treatment? What is the inducement of the initiation of immunosuppression therapy in the patients with MN?

We generally observe patients for 6 months with supportive treatment before considering immunosuppression treatment. However, we may decide to start immunosuppression treatment earlier depending on the clinical circumstances.

We tend to start induction immunosuppression with either Modified Ponticelli or a calcineurin inhibitor for the majority of patients, with a decision usually made at our multi-disciplinary meeting involving nephrologists and pharmacists.

2. The patients are enrolled from 2000 until 2019. During two decades of study period, therapy for PMN was changed substantially. Could the authors compare the remission rate between early period from 2000 to 2010 and late period from 2011 to 2019?

We have run this analysis and included it as supplementary table 2. There was no significant difference in remission (79.3 % in earlier group vs 75.8% in later group, p=0.59) or relapse (45.3% vs 41.8%, p=0.68) rates between the two time periods. More immunosuppression was used in the later period group (60.9% vs 45.8%, p=0.046). 

3. Please show the Kaplan-Mayer analysis for cumulative probabilities of remission against follow-up period.

Figure 3 depicts Kaplan-Meier curves for both overall survival and freedom from RRT for those patients who achieved no, partial or complete remission. If there is further Kaplan-Meier analysis that you were wanting to see then let us know.

4. Deaths was reported in 53 cases out of 178 patients. Approximately 30% of patients died. It is very high if the mortality was caused by kidney diseases. Please identify the cause of death.

As the study period was over 20 years with a median age at presentation of 58.3 years, the mortality data will include non-kidney related death. Unfortunately we were not able to collect specific cause of death data due to the retrospective nature of the study.

Reviewer 4

1. With respect to the ratio of remission or RRT, the ratio is not appropriate, but should be presented as the incidence of RRT with a unit of /patient/yr.

We have kept the rate of remission, relapse, progression to RRT and death as a percentage (number of outcomes observed within the group being described) as we believe this accurately describes these outcomes for the total cohort and various subgroup analyses.

 2. To define the partial and complete remission of proteinuria, please put the appropriate references.

Reference has been included in the appropriate section of the methods section.

3. Please perform the subgroup analysis after blood anti-PLA2R antibody level was started to be assessed routinely even if “our anti-PLA2R data is far from complete”. Or, please specify how incomplete they are. That topic is what not a few nephrologists are so interested in.

We understand the importance of anti PLA2R testing for all patients with membranous nephropathy. Unfortunately we only had 63 anti-PLA2R results available, and of these 34 were positive. As such any subgroup analysis will be difficult to extrapolate to a wider population due to the low numbers involved. Furthermore, when the test was first available only a positive/ negative result was available, but more recently a titre result has been reported, further complicating potential analysis. A prospective study would be a very interesting next step to address these issues.

Minor concerns:

In the abstract MN has been spelled out and the median observation period has been specified.

In the Abstract, the referee does not understand “Amongst the whole cohort, those who went into remission did better than those who did not.” 

This has been clarified in the abstract.

Also, without the definition of high- or low-risk, it is difficult to understand “Those classified as high-risk also had worse outcomes than those at low-risk”.

This has been clarified in the abstract. Note that we included the reference for the classification in the results section (KDIGO 2021 Clinical Practice Guideline for the Management of Glomerular Diseases. Kidney Int. 2021;100:S1-s276.).

---

## [Editor Report · Decision Letter 1]

29 Sep 2022

A low rate of end-stage kidney disease in membranous nephropathy: a single centre study over 2 decades

PONE-D-22-19331R1

Dear Dr. Storrar,

We’re pleased to inform you that your manuscript has been judged scientifically suitable for publication and will be formally accepted for publication once it meets all outstanding technical requirements.

Kind regards,

Donovan Anthony McGrowder, PhD., MA., MSc

Academic Editor

PLOS ONE

Additional Editor Comments:

Dear Dr. Storrar,

 The manuscript was revised in accordance with the reviewers’ comments and is provisionally accepted pending final checks for formatting and technical requirements.

Best regards,

Dr. Donovan McGrowder (Academic Editor)

---

## [Editor Report · Acceptance letter]

4 Oct 2022

PONE-D-22-19331R1 

A low rate of end-stage kidney disease in membranous nephropathy: a single centre study over 2 decades 

Dear Dr. Storrar:

I'm pleased to inform you that your manuscript has been deemed suitable for publication in PLOS ONE. Congratulations! Your manuscript is now with our production department. 

Kind regards, 

on behalf of

Dr. Donovan Anthony McGrowder 

Academic Editor

PLOS ONE